# Determining the Authenticity of *Spirulina* Dietary Supplements Based on Stable Isotope and Elemental Composition

**DOI:** 10.3390/foods12030562

**Published:** 2023-01-27

**Authors:** Jasmina Masten Rutar, Lidija Strojnik, Marijan Nečemer, Luana Bontempo, Nives Ogrinc

**Affiliations:** 1Department of Environmental Sciences, Jožef Stefan Institute, Jamova 39, 1000 Ljubljana, Slovenia; 2Jožef Stefan International Postgraduate School, Jamova 39, 1000 Ljubljana, Slovenia; 3Department of Low and Medium Energy Physics, Jožef Stefan Institute, Jamova 39, 1000 Ljubljana, Slovenia; 4Department of Food Quality and Nutrition, Research and Innovation Centre, Fondazione Edmund Mach, Via Mach 1, 38010 San Michele all’Adige, Italy

**Keywords:** *Spirulina*, *Arthrospira* spp., dietary supplements, stable isotope ratio, elements, authenticity, quality, geographical origin

## Abstract

While the demand for *Spirulina* dietary supplements continues to grow, product inspection in terms of authenticity and safety remains limited. This study used the stable isotope ratios of light elements (C, N, S, H, and O) and the elemental composition to characterize *Spirulina* dietary supplements available on the Slovenian market. Forty-six samples were labelled as originating from the EU (1), non-EU (6), Hawaii (2), Italy (2), Japan (1), Portugal (2), Taiwan (3), India (4), and China (16), and nine products were without a declared origin. Stable isotope ratio median values were –23.9‰ (–26.0 to –21.8‰) for *δ*^13^C, 4.80‰ (1.30–8.02‰) for *δ*^15^N, 11.0‰ (6.79–12.7‰) for *δ*^34^S, –173‰ (– 190 to –158‰) for *δ*^2^H, and 17.2‰ (15.8–18.8‰) for *δ*^18^O. Multivariate statistical analyses achieved a reliable differentiation of Hawaiian, Italian, and Portuguese (100%) samples and a good separation of Chinese samples, while the separation of Indian and Taiwanese samples was less successful, but still notable. The study showed that differences in isotopic and elemental composition are indicative of sample origins, cultivation and processing methods, and environmental conditions such that, when combined, they provide a promising tool for determining the authenticity of *Spirulina* products.

## 1. Introduction

As demand for high-quality food supplements continues to grow, combined with a greater awareness of the importance of food quality and safety, consumers are prioritizing products with declared composition and geographical origin [1,2,3,4,5,6]. There is a limited understanding of nutritional composition across microalgal species, geographical regions, and seasons, all of which can substantially affect quality and safety value of products based on microalgae, which are available on the market, mostly in the form of nutritional supplements [7]. Interest in *Spirulina* dietary supplements is also proliferating due to its accepted nutritional properties, such as high protein, mineral, vitamin, pigment, and other beneficial phytochemicals content [5,8,9]. The high proportion of adulterated algal food products discovered in the past (more than 50%), resulting partly from the lack of quality control measures, reveals the severity of the fraud, which cheats the consumer and poses a potential health risk [10,11].

Microalgal products that are subjected to variations in quality due to unstable culturing conditions are likely to be deliberately adulterated [12]. Common adulterants in microalgae products are flour and mungbean powder, which contain significantly less protein compared to *Spirulina* and *Chlorella* and have lower production costs [5]. Environmental conditions such as climate change and variability that influence algal growth may affect potential biomarkers, but also the stable isotopic composition of sulfur, carbon, and nitrogen. These parameters could be used to verify the origin of microalgae and to detect adulterants in algal products, and can be even more efficient when combined with elemental composition [13]. For example, the changes in isotopic composition of *Spirulina* products are also expected due to different production locations and cultivation techniques used for *Spirulina* production such as open or closed systems. Open systems, while being easier to operate and having lower operating costs, contribute certain negative aspects, such as water loss due to evaporation, carbon loss due to its diffusion to the atmosphere, and changing environment (light intensity, temperature, pH). Additionally, contamination of the *Spirulina* culture with external materials presents a significant problem in this cultivating system [14,15]. In contrast, closed systems are enclosed culture vessels for controlled algal biomass production. They have no direct exchange of gasses or contaminants with the environment. Instead, they provide an environment with controlled water and carbon dioxide supply, temperature, light intensity, and pH regulation, and determined aeration and gas exchange [16]. Additionally, it was found that CO_2_ *δ*^13^C values are lower in closed systems compared to open systems and reflect in lower *δ*^13^C values in algal biomass [17]. 

The information concerning the distribution of the *δ*^34^S values in aquatic organisms and resources is scarce. Generally, there are three potential parameters affecting sulfur isotopic composition in algae: geology, proximity to the sea, and redox chemistry, which can help to provide information about the algae origin. For instance, the *δ*^34^S values of marine sulfate and vegetation near the sea are about +20‰ and continue to decrease over 100 km distance from the sea to +6‰ [18,19]. Further, the hydrogen and oxygen isotopic composition can provide additional information about the region of growing, since they differ according to latitude, altitude, and proximity to the sea [20,21]. After passing through the evaporation, condensation, and precipitation cycle, the meteoric water makes up groundwater with a systematic geographical variation in isotopic composition. Ocean water evaporation causes, through fractionation, a decrease of heavy water isotopomers concentration in the clouds, as compared to the sea. A decrease in the heavy isotope content of the precipitation also occurs due to decreasing temperatures when the equatorial ocean water vapor moves to higher latitudes and altitudes. With clouds moving further inland and gaining altitude, more evaporation, condensation, and precipitation occur, further decreasing the heavy isotope concentrations in water. Accordingly, the isotopic gradient from coastal to inland regions is reflected in the ground waters [22], and finally, the products prepared using this water [20], which, in our case, were *Spirulina* food supplements.

Furthermore, different nutrient mediums are used in *Spirulina* cultivation (Zarrouk medium, wastewater, manure, fresh, and seawater), contributing mainly to different isotopic composition of nitrogen, but also elemental compositions of the *Spirulina* culturing medium and the *Spirulina* biomass [23,24,25,26,27]. The elemental composition of *Spirulina* products has been shown to vary according to changes in the composition of culturing medium and deliberate enrichments of *Spirulina* products, as it has been shown that micronutrient addition in the growth medium notably improves the accumulation of macro- and micronutrients [28,29,30]. Additionally, the pH of the culturing medium has an important impact on mineral assimilation by *Spirulina* biomass, as higher assimilation of metal ions is stimulated at higher pH [31]. However, the cell mineral content grows only to a certain extent, after mineral concentration lowers or stagnates [28]. In addition, commercialization methods, such as processing, packaging techniques and transport, can affect the chemical composition of *Spirulina* products [32]. 

In Slovenia, Kejžar et al. [32] performed a preliminary characterization of algae dietary supplements from Slovenian market, which contributed towards a better overview of quality and authenticity of the products in this market niche. However, this research was restricted to only a limited number of samples, and only a part of analyzed samples were *Spirulina* products. Additionally, only carbon, nitrogen, and sulfur isotopic composition of the tested samples was determined. In the present study, however, all of the available *Spirulina* commercial food supplements from the Slovenian market and more stable isotope parameters were included to achieve a thorough overview of these products in Slovenia. Due to a lack of inspection in this market section, it is important to provide quality control and to raise awareness about quality and origin of these products among consumers. Thus, the objective of this study was to verify the quality and origin of all *Spirulina* dietary supplements sold commercially on the Slovenian market with combining carbon, nitrogen, sulfur and, for the first time, hydrogen and oxygen stable isotope ratios (^13^C/^12^C, ^15^N/^14^N, ^34^S/^32^S, ^2^H/^1^H, ^18^O/^16^O), with elemental composition, to verify the type of production and country of origin. To our knowledge, this is the first time that such an approach has been used to determine the authenticity of *Spirulina* products.

## 2. Materials and Methods

### 2.1. Sample Collection and Preparation

An attempt was made to collect all available *Spirulina* products on the Slovenian market. In total, 46 samples of *Spirulina* food supplements were gathered from physical and online stores over two months in 2018. The majority of samples (44 samples) contained only *Spirulina* spp., while two were mixed samples also containing other plant material (wheat grass and barley grass) or algae (*Chlorella*, *Lithothamnium*). Of the samples containing only *Spirulina* as an active ingredient, the majority (34 samples) were labelled as pure, and ten were declared to contain excipients. The samples were labelled as originating from Hawaii (*n* = 2), Italy (*n* = 2), Japan (*n* = 1), Portugal (*n* = 2), Taiwan (*n* = 3), India (*n* = 4), European Union (EU) (*n* = 1), non-EU (*n* = 6), China (*n* = 16), or were without declared origin (NS; *n* = 9). The samples were sold either dried in tablet, capsule, or powder form, or fresh (Table 1). No additional information was available regarding production and processing. 

Sample preparation included opening the capsules to extract the dry material, grinding the tablets to powder and, in the case of fresh samples, freeze-drying the contents followed by grinding to a powder. Once in powder form, all the samples were kept in sealed plastic containers in the dark at 4 °C.

### 2.2. Stable Isotope Ratio Analysis of Light Elements Using Isotope Ratio Mass Spectrometry

Measurements of the stable isotope ratios of light elements (^2^H/^1^H, ^13^C/^12^C, ^15^N/^14^N, ^18^O/^16^O, ^34^S/^32^S) were performed using Isotope Ratio Mass Spectrometry (IRMS) and are expressed in the *δ*-notation in ‰ according to Equation (1) [33]:(1)δiE=R(Ei/Ej)sample−R(Ei/Ej)standard(Ei/Ej)standard
where *i* stands for the highest, *j* stands for the lowest atomic mass number of the element *E* (H, C, N, O, S), and *R* is the isotope ratio between the heavier and the lighter isotope of the element (^2^H/^1^H, ^13^C/^12^C, ^15^N/^14^N, ^18^O/^16^O, ^34^S/^32^S) in the sample or standard. The *δ*^13^C values are expressed relative to V-PDB (Vienna-Pee Dee Belemnite) standard, *δ*^15^N values relative to AIR, *δ*^34^S values relative to V-CDT (Vienna Cañon Diablo Troilite) standard, and the *δ*^2^H and *δ*^18^O values relative to the VSMOW (Vienna-Standard Mean Ocean Water) standard.

For the stable isotope ratio analysis of light elements C, N, and S, powdered samples (4 mg) and tungsten oxide (WO_3_) (4 mg) were weighed directly into tin capsules which were then sealed and placed into the autosampler of the elemental analyzer. The samples were prepared and analyzed in triplicate. Finally, the mean values were used. The ^13^C/^12^C, ^15^N/^14^N and ^34^S/^32^S values were measured simultaneously by IsoPrime 100 (IsoPrime, Cheadle Hulme, UK)—Vario PYRO Cube (Elementar, Langenselbold, Germany) (OH/CNS Pyrolyser/Elemental Analyzer) with preparation system for solid samples. The following reference materials were analyzed for quality assurance: B2155 (Protein Sercon; *δ*^13^C: −26.98 ± 0.13‰, *δ*^15^N: +5.94 ± 0.08‰, *δ*^34^S: +6.32 ± 0.8‰; CRP (Casein Protein; *δ*^13^C: −20.34 ± 0.09‰, *δ*^15^N: +5.62 ± 0.19‰, *δ*^34^S: +4.18 ± 0.74‰; USGS43 (Indian human hair powder; *δ*^13^C: −21.28 ± 0.10‰, *δ*^15^N: +8.44 ± 0.10‰, *δ*^34^S: +10.46 ± 0.22‰). The measurements’ analytical precision was ± 0.2‰ for *δ*^13^C and *δ*^15^N and 0.3‰ for *δ*^34^S.

The *δ*^2^H and *δ*^18^O measurements were performed at the Fondazione Edmund Mach Research and Innovation Centre, Department of Food Quality and Nutrition (San Michele all’ Adige, Italy). For the analysis of the H and O stable isotope ratios, the *Spirulina* powdered samples were weighed directly into silver capsules (0.20 mg) and then analyzed simultaneously using DELTA XP IRMS (Thermo Scientific, Waltham, MA, USA), coupled with a TC/EA pyrolyzer (Thermo Finnigan, Waltham, MA, USA)). Reference materials applied for normalization of the data were USGS 54 (Pinus contorta, Canadian Lodgepole pine; *δ*^2^H_VSMOW_: –150.40 ± 1.1‰ and *δ*^18^O_VSMOW_: 17.79 ± 0.15‰) and USGS 56 (Berchemia cf. zeyheri, South African red ivory wood; *δ*^2^H_VSMOW_: –44.00 ± 1.8‰ and *δ*^18^O_VSMOW_: 27.23 ± 0.03‰; VSMOW stands for Vienna Standard Mean Ocean Water [34]). The measurements’ analytical precision was ±1‰ for *δ*^2^H and ±0.2‰ for *δ*^18^O.

### 2.3. Macro-Elemental Composition Analysis by X-ray Fluorescence Spectrometry

Analysis of *Spirulina* samples’ macro-elemental composition was performed non-destructively by Energy Dispersive X-Ray Fluorescence Spectrometry (EDXRF) to determine the following elements (13): phosphorous (P), titanium (Ti), zinc (Zn), silicon (Si), bromine (Br), sulfur (S), chlorine (Cl), manganese (Mn), rubidium (Rb), strontium (Sr), potassium (K), calcium (Ca), and iron (Fe). Powdered samples were pressed into 0.5–1.0 g pellets for analysis using a pellet die and a hydraulic press. For fluorescence excitation disc radioisotope, excitation sources Cd-109 (20 mCi, Eckert and Ziegler, Berlin, Germany) and Fe-55 (25 mCi, Eckert and Ziegler, Berlin, Germany) were used. An EDXRF spectrometer with a PX5 digital pulse processor (Amptek, Bedford, MA, USA), an XR-100 SDD detector (Amptek, Bedford, MA, USA), and a PC-based, multichannel analyzer software package (DPPMCA) were used for the emitted fluorescence radiation detection. For light element analysis (Si, P, S, and Cl), the spectrometer operating in Fe-55 mode was equipped with a vacuum chamber, and for K, Ca, Ti, Mn, Fe, Zn, Br, Rb, and Sr analysis, measurements in Cd-109 mode were performed in the air. The energy resolution of the spectrometer was 125 eV at 5.9 keV. AXIL Spectral Analysis software was used to analyze the complex X-ray spectra. For quantification, the Quantitative Analysis of Environmental Samples (QAES) software developed in our laboratory was used [35,36]. Method validation was performed using 1573a (tomato leaves) and 1547 (peach leaves) NIST standard reference materials. The EDXRF analysis estimated uncertainty budget was 11% and was incorporated in the QAES software procedure.

### 2.4. Statistical Analysis

XLSTAT software (Addinsoft, Long Island, NY, USA, 2019) and SIMCA-P (version 17, Sartorius Stedim Biotech, Umeå, Sweden) were used for statistical analysis. Following basic statistical methods (maximum, minimum, median, and quartiles), multivariate statistical analyses methods, including Principal Component Analysis (PCA), Discriminant Analysis (DA), and Orthogonal Partial Least Squares Discriminant Analysis (OPLS-DA), were applied to identify further characteristic parameters for discrimination of samples based on their elemental and isotopic composition. Internal sevenfold cross-validation was used to determine the significant components of the models and thus minimize overfitting. The OPLS-DA study evaluated performance using the explained variation (R^2^X for PCA and R^2^Y for OPLS-DA) and predictive ability (Q^2^). The OPLS-DA model prediction performance was also evaluated via specificity (true negatives) and sensitivity (true positives), calculated as described by Fiamegos et al. [37]. The accuracy (TP/TN)/(TP + FP + FN + TN) and F1 Score (F1 Score = 2 × (Recall × Precision) / Recall + Precision) of the model were calculated as described by Strojnik et al. [38]. Candidates for discriminant markers were selected by loading plots, which allow for visualization of the relationships between the formed groups and the variables and by the variable importance in the projection (VIP) values of the OPLS-DA models, where a value higher than one was considered the threshold. 

## 3. Results and Discussion

### 3.1. Isotopic Composition of Spirulina Food Supplements from the Slovenian Market

While the detailed elemental composition of *Spirulina* dietary supplements from a nutritional point of view has already been presented [9], *Spirulina* stable isotopic profiles, including hydrogen and oxygen isotopic composition, were characterized here for the first time and, together with their macro-elemental composition, were used to verify the country of origin and authenticity of the samples. In the text, the data are presented as the median value (M) and interquartile range (IR, in parentheses) of elemental or isotopic composition. Results of the elemental composition are presented in Appendix A, while stable isotope ratios of light elements C, N, S, O, and H (‰) are collected in Table 2. 

The content of macro-elements (>1 g/kg) in the commercial *Spirulina* supplements was as follows: K > P > S > Si > Cl > Ca, and of micro-elements (>1 mg/kg): Fe > Mn > Sr > Zn > Ti > Br > Rb. Among the macro-elements, potassium values in *Spirulina* samples ranged from 5.83 to 26.9 g/kg with a median value of 15.2 g/kg (IR: 14.3–16.8), phosphorus values ranged from 5.06 to 14.7 g/kg (M: 10.1, IR: 10.1–12.1 g/kg), sulfur from 3.14 to 9.91 g/kg (M: 7.65, IR: 7.14–8.26), silicon from 0.68 to 21.7 g/kg (M: 5.06, IR: 1.46–14.9), chlorine from 0.09 to 5.77 g/kg (M: 2.02, IR: 0.88–2.97), and calcium from 0.46 to 63.5 g/kg (M: 1.56, IR: 0.98–2.82). Among the micro-elements, iron values in *Spirulina* supplements ranged from 0.28 to 3.48 g/kg with a median value of 0.69 g/kg (IR: 0.49–1.13), manganese from 14.7 to 195 mg/kg (M: 33.1, IR: 27.5–46.2), strontium from 4.39 to 478 mg/kg (M: 23.0, IR: 13.1–30.8), zinc from 2.30 to 52.7 mg/kg (M: 15.1, IR: 10.3–21.5), titanium from 2.58 to 65.8 mg/kg (M: 10.4, IR: 5.90–27.3), bromine from 0.47 to 17.4 mg/kg (M: 1.84, IR: 1.25–3.18), and rubidium from 0.50 to 11.9 mg/kg (M: 1.45, IR: 1.11–2.39).

Hawaiian samples (S4 and S26) showed the highest values of Cl, Fe, Zn, Br, and Rb, and the samples declared to originate outside EU had the highest values of Si (S6) and the lowest Cl (S6) and Rb (S16) values. Sample from EU (S18) had the highest values of Ca and Sr, and Italian samples (S44 and S46) had the highest K and the lowest Si, Ca, and Sr (S46) values. The lowest values of P, K, and Fe were found in sample of undeclared origin (S37), while the lowest values of Br were measured in an Indian sample (S7). Chinese samples were the highest in Ti (S23) and S (S43) and the lowest in Zn (S43). Furthermore, the lowest values of Ti and Mn were found in samples of undeclared origin (S28 and S39, respectively). Finally, Portuguese samples (S29, S30) showed the highest Mn values.

The *δ*^13^C values in *Spirulina* commercial samples ranged from –32.3 to –16.7‰, with a median value of −23.9‰ (IR: –26.0 to –21.8‰). The lowest values were measured in Italian (S44, S46) and Indian (S7, S38) samples, the highest in Chinese samples (S33), and those without declared origin (NS; S37). The high *δ*^13^C value of –17.4‰ in the S37 sample could be explained by the presence of corn maltodextrin excipient in the final product (Table 1), which has *δ*^13^C values similar to C_4_ plants (−17 to −9‰), while measured values in *Spirulina* samples are closer to those for C_3_ plants (−40‰ to −20‰) [39]. The same could be assumed for the sample S33, where the presence of undeclared excipient with a higher *δ*^13^C values (such as corn maltodextrin) could explain the high measured *δ*^13^C value of –16.7‰. West et al. [17] found in their research that carbon values higher than −32‰ and lower than −29‰ are characteristic of crops grown in the shade or indoors, and crops with *δ*^13^C values higher than −29‰ were identified as grown outdoors. However, the classification of crops grown outdoors could also include crops grown indoors when good ventilation in the indoor environment was included. Therefore, the classification of an outdoor-grown crop includes open-grown crops cultivated both outside or inside a structure. Sample separation according to *δ*^13^C values is less reliable in our case, as the producers use several different synthetic or organic products to enrich the *Spirulina* growth medium. Therefore, the final products do not reflect the actual *δ*^13^C isotopic composition of the environment and microalgae, but we may assume that algae grown indoor would have lower *δ*^13^C values than −28‰. Our case samples were S1, S7, S34, S38, S40, S44, and S46.

The *δ*^15^N values spanned a broad range of values, i.e., –5.35 to 13.8‰ (M: 4.80‰, IR: 1.30–8.02‰), where the lowest values were again measured in Italian samples (S46, S44) and the highest in Hawaiian (S4, S26), Japanese (S1), and samples without declared origin (S37). A high variability in *δ*^15^N values can be attributed to using organic (manure, wastewater) and inorganic (synthetic) fertilizers. Since the nitrogen source in synthetic fertilizers is atmospheric N_2_, their *δ*^15^N value is around 0‰. In organic fertilizers, the *δ*^15^N values are higher, since they are primarily derived from animal waste [40]. Consequently, the crops fertilized with synthetic fertilizers obtain lower *δ*^15^N values than those fertilized with organic fertilizers. Additionally, a mixture of different fertilizers could be used, resulting in a relatively ambiguous nitrogen isotopic composition. Finally, variability in crop *δ*^15^N value could also result from using different amounts of fertilizer [17,40]. The field and laboratory study performed on algae also indicates that higher *δ*^15^N values (up to 11.1‰) are observed in algae exposed to organic manure compared to those exposed to synthetic inorganic fertilizers [41]. Another explanation for high *δ*^15^N values could be the use of a pool of NH_4_^+^ enriched in ^15^N. For instance, in Delaware estuary, the *δ*^15^N values of seston reached a maximum of +18‰ due to the fractionation during assimilation of NH_4_^+^ ions [42].

The *δ*^34^S values ranged from –1.75 to 13.8‰ (M: 11.0‰, IR: 6.79–12.7‰). Here, the lowest values belonged to Indian (S7) and Italian (S44) samples, while the highest values were measured in Chinese (S11, S23, S25) and NS (S20, S42) samples. Studies on algae have shown little isotopic discrimination during the assimilation and reduction of sulfate. The isotopic composition of total sulfur in algae is depleted in ^34^S by only 1–2‰ regarding the dissolved sulfate, which indicates that algae sulfate metabolism involves little or no isotope fractionation [43]. Therefore, *δ*^34^S values in algae will reflect those of meteoric water or water used in their growth medium and geology [44,45]. It is interesting to note that in the Hawaiian Islands, *δ*^34^S values of sulfates from volcanic ash and basalt-derived soils range from 6.3 to 15.4‰ [46] that are also in agreement with our data.

The lowest *δ*^2^H was determined in the NS (S39, S45), Chinese (S11, S27), and sample S5 (outside EU), while the highest values were found in NS samples with undeclared origins (S22, S37). The *δ*^2^H values ranged from –207 to –97.4‰ (M: –173‰; IR: –190 to –158‰). Finally, the *δ*^18^O values ranged from 12.8 to 27.2‰ (M: 17.2‰; IR: 15.8–18.8‰), with examples of NS (S2, S22, S37) and Hawaiian (S4, S26) samples possessing the highest values and Chinese (S11) and certain NS (S28, S45) samples possessing the lowest.

The data also show a good linear correlation between *δ*^2^H and *δ*^18^O (Figure 1), with the slope (y = 7.8x−308.8; r^2^ = 0.82, *p* < 0.001) being comparable to the Global Meteoric Water Line (GMWL). 

The GMWL defines the ratio of the stable isotopes in natural meteoric waters (i.e., water derived from snow, rain, and other forms of precipitation) and is typically defined by the following equation: *δ*^2^H = 8.2 × *δ*^18^O + 10.8 [22]. Thus, our data indicate that hydrogen and oxygen isotopes in analyzed *Spirulina* samples originate mainly from local meteoric water and are only minimally affected by other processes such as metabolism.

### 3.2. Geographical Discrimination of Spirulina Samples from the Slovenian Market

#### 3.2.1. Principal Component Analysis of All *Spirulina* Samples

Principal Component Analysis (PCA) was applied to identify trends and examine the distribution of variables in the analyzed samples. For this analysis 46 *Spirulina* commercial samples from the Slovenian market were obtained, and 18 analyzed parameters (macro-elemental and isotopic composition data) were used (Figure 2). In the PCA score plot (Figure 2a), a grouping of the samples can be observed corresponding to different elemental and isotopic compositions of the included samples. Here, three outstanding groups represented by different variables can be identified. Information about the variables that contributed most to the grouping of the samples in the PCA is provided in the PCA variables loading plot (Figure 2b).

The first group of outliers (S37 and S22; Figure 2a) is characterized by having lower levels of P (S37: 6.16 and S22: 6.82 g/kg; M: 10.9 g/kg, IR: 10.1–12.1 g/kg), K (S37: 5.83 and S22: 7.40 g/kg; M: 15.2 g/kg, IR: 14.2–16.8 g/kg), Fe (S37: 0.28 and S22: 0.39 g/kg; M: 0.69 g/kg, IR: 0.49–1.13 g/kg), and S (S37: 3.88 and S22: 3.60 g/kg; M: 7.65 g/kg, IR: 7.14–8.26 g/kg). The lower content of these elements suggests a lower *Spirulina* content representing possible adulteration, since *Spirulina* typically contains high levels of P, K, Fe, and Zn [9,47,48]. Alternatively, a different growth medium could also result in different mineral compositions. For example, Michael et al. [49] showed a connection between using a poorer culturing medium (regarding elemental composition) and a lower mineral content in the final *Spirulina* product. However, given that S37 had the second highest *δ*^15^N value (13.3‰) among all samples, the use of organic fertilizers in its cultivation can be suggested. As this type of cultivation medium is rich in nutrients and has a positive effect on *Spirulina* mineral uptake [25,26,27], it is unlikely for the growth medium to be responsible for the poor mineral content in this sample. Additionally, the highest *δ*^18^O (S37: 25.8‰, S22: 27.2‰) and *δ*^2^H values (S37: –105‰, S22: –97.4‰) among all samples were observed for S37 and S22 (Figure 2b). High *δ*^18^O and *δ*^2^H values could be attributed to the proximity of the *Spirulina* culturing site to the equatorial region and the short distance from the sea, as hydrogen and oxygen isotopic composition are strongly latitude- and altitude-dependent [20,22].

Among the samples within the elliptical field in Figure 2a, samples S18, S9, and S44 in the upper right quadrant appear to form a separate group with distinct characteristics. Here, S18 and S44 are differentiated by their high Sr values (478 and 86.2 mg/kg, respectively; M: 23.0 mg/kg, IR: 13.1–30.8 mg/kg) and low *δ*^15^N values (0.77 and –3.92‰, respectively) compared to other samples (median value for *δ*^15^N: 4.80‰, IR: 1.30–8.02‰). Additionally, S18 also has the highest Ca content. High Ca and Sr values in S18 can be explained by the algae *Lithothamnium* in this sample. In contrast to *Spirulina* and *Chlorella*, which also make up this product and contain moderate Ca and Sr levels, *Lithothamnium* algae contain higher values of these elements [50,51]. In S44, the higher level of Sr could be attributed to contamination during the flaking process, as this is the only sample in a flake form. The flaking process’s impact on contamination with certain elements has been shown several times in previous research, where metal contamination came from the enameled parts of the flaking rollers [52,53]. As *δ*^15^N values are primarily influenced by the nitrogen isotopic composition of the nitrogen source used during cultivation and internal transformations, they can indicate the type of fertilizer used, e.g., lower *δ*^15^N values, as are observed in S18 and S44, point to the use of inorganic fertilizers [17,40,54].

Additionally, S9 and S44 possess lower P content (S9: 5.06 and S44: 8.56 g/kg) than most samples (M: 10.9 g/kg, IR: 10.1–12.1 g/kg). This could be explained by the samples’ lower amount of algal material since this is a mixed product, containing, in addition to *Spirulina*, *Chlorella*, barley and wheat grass. While algae (in this case, *Spirulina* and *Chlorella*) are rich in phosphorus, this is not true for cereal grasses, whose content is lower [47,48]. The lower P content in S44 (and partially also S9) could be due to the drying technique used in its production. It has been shown that different drying techniques reduce the P levels in the dried material compared to the fresh one [55]. As can be observed in Figure 2b, all the samples in this group (S9, S18, and S44) have somewhat higher oxygen and hydrogen stable isotope ratios in common, which range from 19.9 to 21.9‰ (M: 17.2‰, IR: 15.8–18.8‰) and from –146 to –128‰ (M: –173‰, IR: –190 to –158‰), respectively.

Rubidium (Rb), Br, Fe, Zn, and Cl appear to play an essential role (due to their high value) in the grouping of the Hawaiian samples (S26 and S4), as well as S2, which are outlying in the lower right quadrant (Figure 2a). Here, the Rb values are as follows: S26: 9.96 mg/kg, S4: 11.9 mg/kg, and S2: 7.47 mg/kg (Rb median value (M) for *Spirulina* samples: 1.45 mg/kg, interquartile range (IR): 1.11–2.39 mg/kg). The Br values for S26 are 17.4 mg/kg, for S4 16.5 mg/kg, and for S2 11.2 mg/kg (M: 1.84 mg/kg, IR: 1.25–3.18 mg/kg). The Fe content in S26, S4, and S2 was 3.09, 3.48, and 3.29 g/kg, respectively, with a median of 0.69 g/kg (IR: 0.49–1.13 g/kg), while the Zn values in S26, S4, and S2 were 35.5, 52.7, and 43.6 mg/kg, respectively, and the median value for all samples was 15.1 mg/kg (IR: 10.3–21.5 mg/kg). The Cl content was 5.63 g/kg for S26, 5.77 g/kg for S4, and 3.07 g/kg for S2 (M: 2.02 g/kg, IR: 0.88– 2.97 g/kg). The higher content of these elements in the Hawaiian and S2 samples may be due to their deliberate addition to the growth medium, which enables *Spirulina* to uptake and accumulate these elements. *Spirulina* elemental content has been previously shown to reflect that of the culturing medium [28,30,31]. Additionally, mineral addition to the *Spirulina* culturing medium to enhance its efficiency as a nutritional source of various minerals is not uncommon [28,30,31,56]. Additionally, the high content of elements Rb, Fe, Zn, and Cl could result from manure used as a fertilizer in the growth medium, as it has been shown that adding manure in crop cultivation results in elemental composition enhancement of the plants [57,58]. A deviation in *δ*^18^O and *δ*^2^H has also been observed in Hawaiian samples (S4, S26) and S2, where *δ*^18^O values ranged from 21.1 (S26) to 21.6‰ (S4) and *δ*^2^H values from –141 (S2) to –136‰ (S26) (median values for *δ*^18^O: 17.2‰ (IR: 15.8–18.8‰) and *δ*^2^H: –173‰, (IR: –190 to –158‰)). Similarly, as previously shown in samples S22 and S37, high *δ*^18^O and *δ*^2^H values can be attributed to the proximity of the production site to the equatorial region and sea and production at low altitudes [20,22]; all parameters are true for Hawaii. However, Figure 3 shows how samples S22 and S37 cannot be placed under Hawaiian samples due to differences in elemental composition. Contrarily, these samples appear to coincide with samples from non-EU regions or Asia, as the content of elements Cl, Fe, Zn, Br, and Rb (Figure 3b–f) is similar.

Another parameter, common to S4, S26, and S2 samples, is the high *δ*^15^N values. Measured values are 10.8‰, 13.8‰, and 8.81‰, respectively, with a median value for all samples of 4.80‰ (IR: 1.30–8.02‰). The nitrogen isotopic composition of the samples provides information about regional agricultural practices [20]. High *δ*^15^N values indicate the use of organic fertilizers, as the manure’s stable isotopic composition has higher *δ*^15^N values than mineral fertilizers [59]. While Zarrouk’s medium (Appendix A) [60] is widely used as a standard medium for *Spirulina* production, manure (chicken, cow, pig) in *Spirulina* cultivation is also a common practice, as it represents a low-cost source of nitrogen and other necessary nutrients. The use of manure in *Spirulina* production results in good cellular growth and high pigment content while reducing production costs [25,26,27]. Additionally, the organic manures are enriched with microflora, which induces crops to uptake micronutrients [56]. The latter can be seen in our study in the high content of specific elements in S4, S26, and S2, as mentioned earlier.

Looking at these results, we can see a close connection between the Hawaiian samples, S4 and S26, and the sample without specified country of origin, S2. Figure 3 presents box plots of *δ*^15^N, Cl, Fe, Zn, Br, and Rb composition of *Spirulina* food supplements according to declared country of origin. The samples of undeclared origin (NS) are marked to compare their isotopic or elemental values with those with declared origins. Sample S2 shows the highest values of Fe, Zn, Br, and Rb among the NS samples. Fe and Zn values fall in the range of the values measured in Hawaiian samples (Figure 3c,d), indicating that it might also originate from Hawaii.

The clustering of the Italian samples (S44 and S46) using PCA analysis was unsuccessful (Figure 2a), despite having many common characteristics. One of the parameters separating them is the high Sr value in the S44, which was sold as flakes. As mentioned earlier, this observation is believed to be due to contamination arising during the flaking process. Unlike S44, S46 was obtained fresh and was subsequently lyophilized. Unlike the flaking process, lyophilization does not cause contamination and has little effect on mineral loss, except in the case of Mn [52,53,61]. Sample S44 has higher Ca content (3.45 g/kg), while the content in S46 is lower (0.46 g/kg). This finding could result from different processing, drying, and flaking techniques and undeclared excipients, as shown in previous research [55,62,63]. Additionally, the drying technique used can explain the higher level of Mn in S44 (84.9 mg/kg) than in S46 (32.9 mg/kg). As previously shown, freeze-drying can cause a decrease in the Mn content compared to oven drying [55]. The Zn content was also higher in the flaked sample (24.9 mg/kg) than in S46 (7.59 mg/kg), which is again believed to be a result of the flaking process [63] or selected drying method, as different drying techniques affect the Zn content differently [55]. Substantial variations in mineral profile have been observed between natural *Spirulina* and commercial products by Campanella et al. [62], which could result from various changing parameters introduced by the commercialization of the product, such as *Spirulina* biomass treatment, processing (washing, drying), packaging, and distribution.

#### 3.2.2. Discriminant Analysis of Spirulina Samples

Discriminant analysis (DA) was performed using macro-elemental and isotopic composition data (Figure 4) to investigate the distribution of variables in more detail and reveal possible differences among the samples originating from China (*n* = 16), Hawaii (*n* = 2), India (*n* = 4), Italy (*n* = 2), Portugal (*n* = 2), and Taiwan (*n* = 3). The first two discriminant components (F1 and F2) account for 90.3% of the total variance. In the discriminant function score plot (Figure 4a), each cluster (centroid) is represented by a scatter plot. In the loadings plot (Figure 4b), they appear as vectors demonstrating a degree of association of the corresponding initial variable with the first two discriminant components. Red vectors indicate the most significant variables, and blue vectors represent the least significant variables for sample separation and clustering. Six groups of samples are identified in the DA score plot (Figure 4a).

A leave-one-out cross-validation (LOOCV) classified 82.8% of the samples correctly. The prediction ability was the highest for Hawaii, Italy, and Portugal (100%), and was the lowest for Taiwan (66.7%). The most critical variables for sample separation are Fe, Br, K, and P. DA analysis confirms our previous findings regarding distinct elemental and isotopic composition of Hawaiian samples. In addition, the separation of Italian and Portuguese samples was achieved using DA (Figure 4a). 

Despite previously presented differences, the Italian samples have several similar characteristics which separate them from other samples in this study. For example, they possess the lowest content of Si (S44: 0.78 g/kg, S46: 0.68 g/kg; M: 5.06 g/kg, IR: 1.46–14.9 g/kg) and one of the lowest values of P (S44: 8.56 g/kg, S46: 6.64 g/kg; M: 10.9 g/kg, IR: 10.1–12.1 g/kg), the highest content of K (S44: 20.6 g/kg, S46: 26.9 g/kg; M: 15.2 g/kg, IR: 14.2–16.8 g/kg) among all samples, and a high Br content (S44: 8.00 mg/kg, S46: 7.07 mg/kg; M: 1.84 mg/kg, IR: 1.25–3.18 mg/kg). Similarly, as with samples from Hawaii, such an elemental composition could reflect the growth medium used, which appears to be specific for this production region. Moreover, S44 and S46 have the lowest *δ*^15^N values (–3.92 and –5.35‰, respectively), which points to the absence of organic fertilizers and the possible influence of natural processes such as nitrification, providing them with sufficiently specific *δ*^15^N composition that distinguishes these samples from others. The Italian samples also have one of the lowest *δ*^34^S values (S44: –0.61‰ and S46: 0.94‰; M: 11.0‰, IR: 6.79–12.7‰), which could be a result of combined meteoric water *δ*^34^S value and that of the added sulfur compounds in the *Spirulina* culturing medium. It could also indicate that *Spirulina* in S44 and S46 was cultivated in freshwater, since the *δ*^34^S values in seaweed are closer to seawater values, i.e., 17 to 21‰ [64,65].

Commercial *Spirulina* samples of Portuguese-declared origin appear to possess a distinct Mn and Br composition (Figure 4a,b). The Mn values for S29 and S30 are 192 and 195 mg/kg, respectively, with a median value for samples analyzed in DA (M_All_) of 33.3 mg/kg (IR_All_: 28.0–38.2 mg/kg). The Br values in these samples are 2.71 mg/kg for S29 and 3.21 mg/kg for S30 (M_All_: 1.62 mg/kg, IR_All_: 1.04–3.21 mg/kg). Additionally, Portuguese samples have the highest Fe content, i.e., 1.14 (S29) and 1.12 g/kg (S30) (M_All_: 0.72 g/kg, IR_All_: 0.56–1.39 g/kg) and Si content (S29: 15.6 g/kg, S30: 15.1 g/kg; M_All_: 7.56 g/kg, IR_All_: 1.43–15.1 g/kg). Mn is the most important parameter for separating Portuguese and Chinese samples, together with *δ*^34^S values (S29: 7.34‰, S30: 7.01‰). The Mn, Br, and Fe levels in *Spirulina* products are highly dependent on their concentration in the growth medium; therefore, adding these minerals will result in increased levels in the microalgae. Moreover, the uptake of these elements is strongly affected by growth conditions, i.e., light intensity [28,30,31,66,67]. In this respect, Portuguese *Spirulina* product production might be specific. Regarding the Mn and Br content, using rich wastewater in these elements also increases their concentration in *Spirulina* [58]. High Si content results from silicon dioxide excipient’s addition to the final product, which is evident from the declared product content (Table 1). The *δ*^34^S composition of Portuguese *Spirulina* samples is in agreement with local *δ*^34^S results for rainwater (*δ*^34^S: 7.2‰) measured in neighboring Spain [68].

Further, OPLS-DA was applied to analyze the observed differences between six countries in the DA analysis and to investigate the goodness of fit (R^2^X) and prediction (Q^2^) for the model. Obtained OPLS-DA resulted in five predictive and no orthogonal components (5 + 0), producing an R^2^X = 0.73, R^2^Y = 0.68, and Q^2^ = 0.48. The F1 Score rate obtained by internal cross-validation was 86.2%, sensitivity was 96.2%, specificity was 57.1%, and accuracy was 87.9%. This model displayed high quality and goodness of fit and predictability (≥0.93) to differentiate among different countries, with the exception of Taiwan, where we obtained 0% predictability, which also supports our DA model. Moreover, OPLS-DA analysis for pairwise comparisons among all six countries (Figure 5) was calculated, similarly as in the study performed by Potočnik et al. [69]. The separation between classes in the OPLS-DA score plots is evident. In Figure 5, separation of the most numerous class (China) from Hawaii, India, Italy, Portugal, and Taiwan is presented, while additional pairwise comparisons are presented in Appendix A. The prominent factors influencing OPLS-DA models were *δ*^18^O, Rb, Br, Fe, *δ*^2^H, Mn, Zn, K, and Cl. These results can be used to delve deeper into discussion regarding distinction of *Spirulina* sample groups, more precisely, Chinese, Indian, and Taiwanese groups.

The *Spirulina* samples of Chinese origin form the strongest group, with good discrimination results. The most critical variables for separation of this group (Figure 4a) are *δ*^34^S, Sr, Si, and *δ*^15^N. Specifically *δ*^34^S values are high in these samples with the median value (M_Ch_) of 12.9‰, which is higher than the median of all samples (11.0‰, IR_All_: 6.72–12.9‰) included in the DA (M_All_). High *δ*^34^S values could result from combined rainwater *δ*^34^S values from the production area and the addition of sulfur compounds in the *Spirulina* growth medium. Several studies have shown that rainwater in various Chinese regions contains a wide range of *δ*^34^S values (0.30‰ to 19.1‰), with higher values attributed mainly to high atmospheric pollution from coal burning in the area. Additionally, heavier *δ*^34^S values were observed in the winter, when coal burning from southern China is the dominant source of pollution [70,71,72]. Air pollution could, therefore, also be reflected in *Spirulina* products produced in polluted areas. The Sr values are also the highest among these samples, M_Ch_ for Sr is 28.1 mg/kg (M_All_: 26.1, IR_All_: 15.2–31.1 mg/kg), which could be attributed to specific *Spirulina* processing techniques used [52,53]. In contrast, Chinese samples contain the lowest amount of Si (M_Ch_: 1.71 g/kg) among the samples included in the DA (M_All_: 7.56 g/kg, IR_All_: 1.43–15.1 g/kg), and the lowest *δ*^15^N value (M_Ch_: 2.32‰; M_All_: 3.04‰, IR_All_: 0.95 to 6.44‰). Lower Si content could partially be explained by the declared pure *Spirulina* composition of Chinese samples, while all Portuguese and half of the Indian samples contain Si as a part of the silicon dioxide excipient. Additionally, higher Si content in non-Chinese samples could come from various *Spirulina* growth medium mineral additions and Si in the water used for cultivation and different processing techniques. Different food processing techniques are known to reduce the Si content in the final product [73].

Results presented in Figure 5a–d confirm the results presented by PCA and DA analyses regarding distinct composition of Hawaiian, Italian, and Portuguese samples. Additional pairwise comparisons to confirm these findings are available in Appendix A.

In addition, Figure 5b,e offer a closer look into separation of Chinese, Indian, and Taiwanese samples, where separation by DA was less successful. In Taiwanese samples, the highest *δ*^13^C values were determined among samples analyzed with DA (M_Taiw_: –21.8‰), while Chinese (M_Ch_: –23.1‰) and Indian (M_Ind_: –28.8‰) samples possess lower *δ*^13^C values (M_All_: –23.9‰, IR_All_: –26.6 to –22.18‰). Differences in *δ*^13^C composition could be due to the addition of different nutrients to the growth medium, excipients to the final product, and cultivation conditions (open or closed system). Closed production systems enable better control over cultivation conditions, such as loss of CO_2_ to the atmosphere, temperature, and pH, and possess lower carbon dioxide *δ*^13^C values [17,74]. There is a considerable difference in the *δ*^34^S values among Indian, Taiwanese, and Chinese samples as well—the *δ*^34^S values in Indian samples are substantially lower (5.7‰) than in Taiwanese samples (11.5‰) and Chinese samples (12.9‰). As Taiwan is an island, a higher influence of the sea than on the mainland (India, China) is expected. The *δ*^34^S values in rainwater decrease while moving inland and further from the sea. Therefore, higher *δ*^34^S values in rainwater and plants and algae are expected in regions closer to the sea [75]. In addition, Taiwan is a volcanic island, and therefore higher *δ*^34^S values can be also expected [46,76]. Higher *δ*^34^S values in Chinese samples, on the other hand, could be attributed, as mentioned previously, to high pollution levels in the area [70,71,72]. Separation of the Indian and Taiwanese samples from Chinese samples is also due to higher *δ*^15^N values (M_Ch_: 2.32‰, M_Ind_: 6.13‰, M_Taiw_: 6.22‰), which points to a specific fertilizing technique for these areas which possibly includes the use of organic fertilizers. Important parameters for distinction between Chinese, Taiwanese, and Indian samples are also *δ*^18^O and *δ*^2^H values, which are the highest in Taiwanese (*δ*^18^O: 18.8‰, *δ*^2^H: –171‰) and Indian samples (*δ*^18^O: 18.7‰, *δ*^2^H: –168‰), while the values in Chinese samples are lower (*δ*^18^O: 16.8‰, *δ*^2^H: –179‰). This could be attributed to the production of Taiwanese and Indian samples at lower latitudes, altitudes, or closer to the sea [20,22], similarly to Hawaiian samples. Another important variable that separates *Spirulina* samples from China, Taiwan, and India is the Si content, the median of which for Indian samples (M_Ind_) is 13.8 g/kg, for Taiwanese samples (M_Taiw_) is 7.94 g/kg, and for Chinese samples (M_Ch_) is 1.71 g/kg (M_All_: 7.56, IR_All_: 1.43–15.1 g/kg), which can be explained by SiO_2_’s addition to the final product in the case of Indian samples. Taiwanese and Chinese *Spirulina*, on the other hand, are declared as pure. As described above, there are various parameters affecting the Si content in these products, such as addition during cultivation, varying water Si concentration, and different processing techniques [73]. Differences were also found in Zn (M_Taiw_: 15.7 mg/kg, M_Ch_: 15.8 mg/kg, M_Ind_: 10.9 mg/kg), Fe (M_Taiw_: 0.66 g/kg, M_Ch_: 0.77 g/kg, M_Ind_: 0.46 g/kg), and K (M_Taiw_: 13.6 g/kg, M_Ch_: 15.6 g/kg, M_Ind_: 15.1 g/kg) values, which could be a result of mineral addition and use of different fertilizers in *Spirulina* growth medium [28,30,31,57,58].

According to the presented data, reliable and specific classification of samples S20, S21, S28, S39, S42, and S45 of undeclared origin (NS) is not possible. However, their elemental and isotopic compositions show similarity with Asian and declared non-EU samples (Figure 3). The findings of this study show the importance of combining both elemental and isotopic values in verifying the country of origin and authenticity of *Spirulina* samples. However, the prediction ability and assessment of authenticity should be improved in the future by including a higher number of samples, as well as verified pure *Spirulina* samples.

## 4. Conclusions

Interest in *Spirulina* dietary supplements is growing among consumers due to vegetarianism, increasing malnutrition, and health awareness in the population. The high demand for *Spirulina* products and the challenging culturing conditions needed for producing high-quality products make them a target for intentional adulteration and mislabeling [12,77]. This study has shown that combining stable isotope ratios of light elements (C, N, S, H and O) and elemental composition creates a promising tool for determining the authenticity of the commercial *Spirulina* dietary supplements, regarding their composition and geographical origin. Hydrogen and oxygen stable isotope ratios have been determined in *Spirulina*-based products for the first time in this study and show a correlation, as it occurs also in water, indicating that they originate mainly from local precipitation and that the influence of other parameters on their values is negligible.

A wide variability in the stable isotopic ratios and elemental composition among *Spirulina* samples of different declared origins was observed. Different statistical methods and reliable discrimination of Hawaiian, Italian, and Portuguese samples were also achieved, together with a good separation of Chinese samples. Discrimination between Taiwanese and Indian samples, however, was less successful but still notable. The parameters responsible for sample discrimination appear to be different culturing and processing techniques, environmental conditions (including pollution), and the geographical location of *Spirulina* production.

Additionally, this method shows promising results in exposing adulterated samples and samples mixed with other products and could be used in future studies for assessing product authenticity. A higher number of samples and more precise information on product culturing conditions, composition, and origin would result in more reliable results in *Spirulina* product authentication.

## Figures and Tables

**Figure 1 foods-12-00562-f001:**
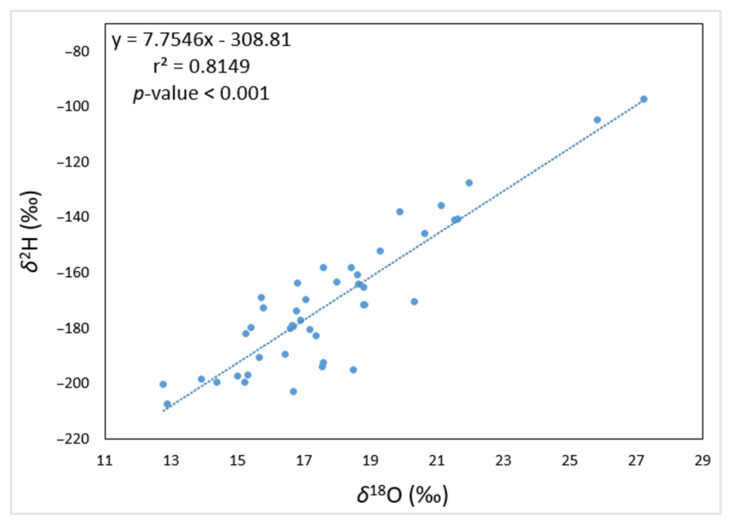
A plot of *δ*^2^H and *δ*^18^O relationship in *Spirulina* food supplements. The blue dots in the plot represent the analyzed *Spirulina* samples and the dotted line through the data points indicates a correlation of the data (y = 7.8x − 308.8; r^2^ = 0.82, *p* < 0.001).

**Figure 2 foods-12-00562-f002:**
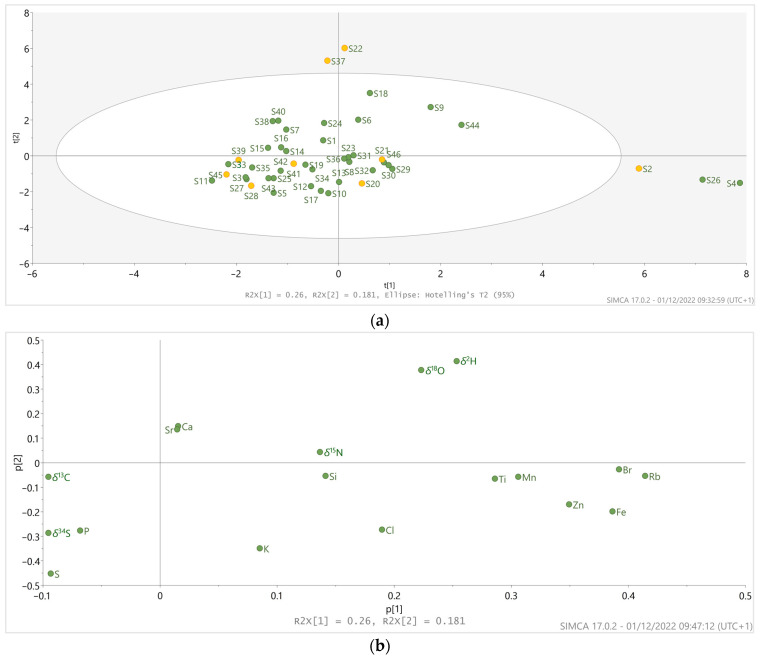
(**a**) PCA score plot of *Spirulina* food supplements (*n* = 46) available on the Slovenian market and (**b**) PCA variables loading plot. Orange circles in figure (**a**) mark the samples of undeclared origin (NS).

**Figure 3 foods-12-00562-f003:**
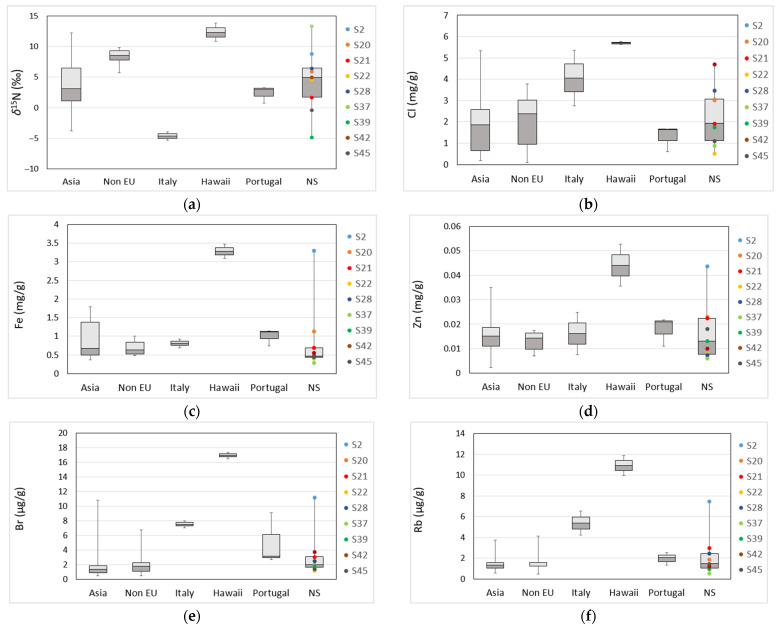
Box plots of selected elemental and nitrogen isotopic composition of *Spirulina* food supplements according to declared geographical origin, with specifically marked samples of undeclared origin (NS). (**a**) *δ*^15^N, (**b**) Cl, (**c**) Fe, (**d**) Zn, (**e**) Br, and (**f**) Rb. The values presented under the label ‘Asia’ include Japanese, Indian, Chinese, and Taiwanese samples.

**Figure 4 foods-12-00562-f004:**
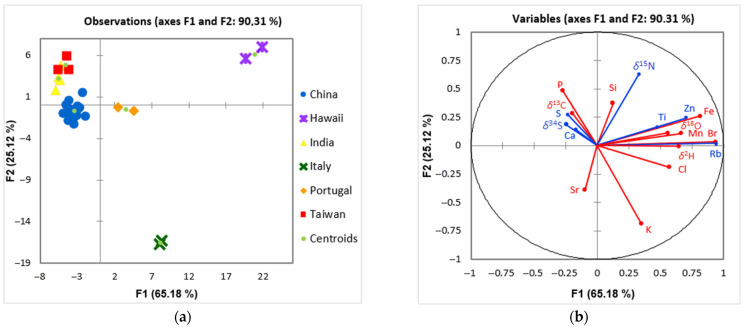
Discriminant function score plot (**a**) and discriminant loadings plot (**b**) for *Spirulina* food supplements available on the Slovenian market (China (*n* = 16), Hawaii (*n* = 2), India (*n* = 4), Italy (*n* = 2), Portugal (*n* = 2), and Taiwan (*n* = 3)). Red vectors indicate the most, and blue vectors are the least significant variables for sample separation.

**Figure 5 foods-12-00562-f005:**
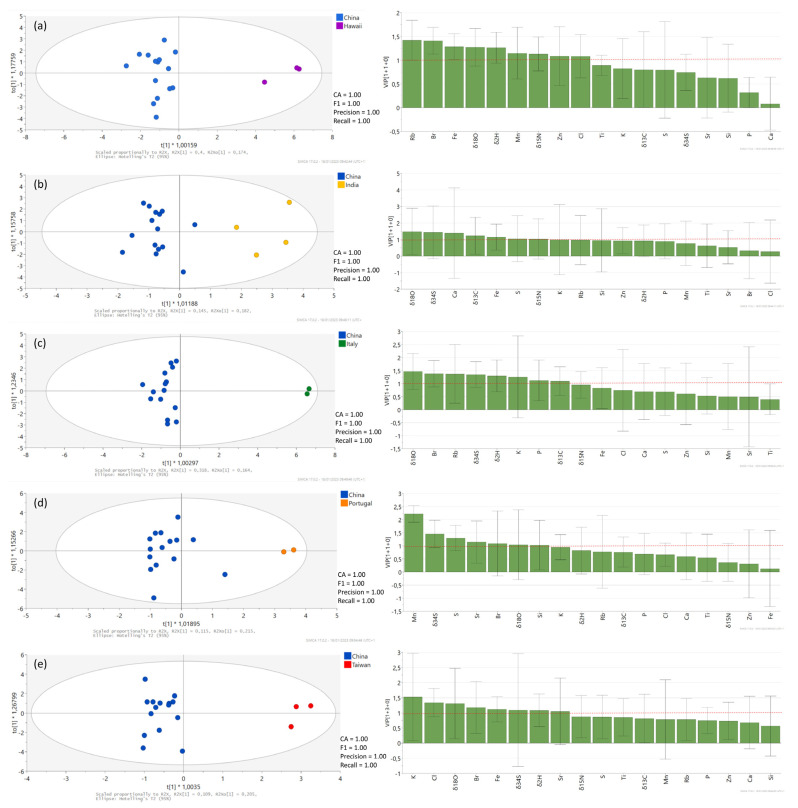
OPLS-DA score plots and VIP values in the pairwise comparisons between different declared countries of origin of *Spirulina* products derived from all isotopic and elemental composition data. The ellipse on the score plot represents the 95% confidence interval. Red dotted line indicates a criteria for identification of the variables, important for the developed model. Separation of the most numerous class, China, is presented, from Hawaii (**a**), India (**b**), Italy (**c**), Portugal (**d**), and Taiwan (**e**).

**Table 1 foods-12-00562-t001:** List of collected pure *Spirulina* or mixed supplements on the Slovenian market with product content, form, and geographical origin, as declared on the label.

Sample	Declared Origin	Product Content	Form
S1	Japan	*Spirulina*, edible scallop shell powder, edible refined processing fat	Tablets
S2	NS	*Spirulina pacifica*, Mg(C_18_H_35_O_2_)_2_	Capsules
S3	Outside EU	*Spirulina platensis*	Powder
S4	Hawaii	*Spirulina pacifica*, SiO_2_, chicory inulin, Mg(C_18_H_35_O_2_)_2_	Tablets
S5	Outside EU	*Spirulina*	Powder
S6	Outside EU	*Spirulina platensis*	Powder
S7	India	*Spirulina platensis*	Tablets
S8	China	*Spirulina platensis*	Tablets
S9	Mongolia–China	Wheatgrass, Barley grass, *Spirulina*, *Chlorella*	Powder
S10	China	*Spirulina*	Powder
S11	China	*Spirulina platensis*	Powder
S12	China	*Spirulina*	Powder
S13	China	*Spirulina*	Tablets
S14	Taiwan	*Spirulina platensis*	Tablets
S15	Taiwan	*Spirulina platensis*	Powder
S16	Outside EU	*Spirulina*	Powder
S17	China	*Spirulina platensis*	Powder
S18	EU	*Spirulina*, *Chlorella*, *Lithothamnium*	Capsules
S19	India	*Spirulina platensis*, CaCO_3_, micro-crystalline cellulose, stearic acid, CMC, SiO_2_	Tablets
S20	NS	*Spirulina*, SiO_2_, Mg(C_18_H_35_O_2_)_2_	Tablets
S21	NS	*Spirulina*	Tablets
S22	NS	*Spirulina platensis*, maltodextrine, SiO_2_, Mg(C_18_H_35_O_2_)_2_, HPMC	Tablets
S23	China	*Spirulina*	Tablets
S24	China	*Spirulina*	Powder
S25	China	*Spirulina*	Tablets
S26	Hawaii	*Spirulina pacifica*, SiO_2_	Tablets
S27	China	*Spirulina*	Powder
S28	NS	*Spirulina*	Tablets
S29	Portugal	*Spirulina platensis*, SiO_2_, Mg(C_18_H_35_O_2_)_2_	Tablets
S30	Portugal	*Spirulina platensis*, SiO_2_, Mg(C_18_H_35_O_2_)_2_	Tablets
S31	China	*Spirulina platensis*	Powder
S32	China	*Spirulina*	Tablets
S33	China	*Spirulina*	Powder
S34	India	*Spirulina platensis*	Tablets
S35	Outside EU	*Spirulina*	Tablets
S36	Outside EU	*Spirulina*	Powder
S37	NS	*Spirulina maxima*, corn maltodextrin, Mg(C_18_H_35_O_2_)_2_	Tablets
S38	India	*Spirulina platensis*	Tablets
S39	NS	*Spirulina*	Tablets
S40	Taiwan	*Spirulina platensis*	Tablets
S41	China	*Spirulina platensis*	Powder
S42	NS	*Spirulina*	Capsules
S43	China	*Spirulina*	Powder
S44	Italy	*Spirulina platensis*	Flakes
S45	NS	*Spirulina*	Powder
S46	Italy	*Spirulina platensis*	Fresh

NS—Not Specified; CMC—Croscaramellose sodium; HPMC—Hydroxypropyl methyl cellulose.

**Table 2 foods-12-00562-t002:** Stable isotope ratios of light elements C, H, S, O, and N (‰) in *Spirulina* dietary supplements sold on the Slovenian market.

Sample Number	^13^C/^12^C, ^2^H/^1^H, ^34^S/^32^S, ^18^O/^16^O, ^15^N/^14^N Isotope Ratio Expressed in *δ*-Notation (‰)				
	*δ*^13^C	*δ*^2^H	*δ*^34^S	*δ*^18^O	*δ*^15^N
S1	−28.0	−152	12.3	19.3	12.2
S2	−25.1	−141	7.46	21.5	8.81
S3	−20.4	−197	11.8	15.3	5.72
S4	−25.8	−141	8.78	21.6	10.8
S5	−26.1	−203	11.3	16.7	8.81
S6	−22.1	−119	−0.60	17.1	7.61
S7	−29.6	−165	−1.75	18.8	8.62
S8	−25.1	−163	13.4	18.0	1.16
S9	−26.6	−146	6.72	20.6	5.59
S10	−22.9	−177	12.8	16.9	6.44
S11	−26.1	−200	13.8	12.8	2.31
S12	−22.3	−179	12.9	16.6	−0.88
S13	−22.7	−179	12.9	16.7	−1.97
S14	−21.8	−172	11.5	18.8	6.22
S15	−21.7	−171	11.7	18.8	6.56
S16	−27.9	−164	2.67	16.8	9.84
S17	−22.2	−180	13.4	16.6	−2.40
S18	−18.1	−138	10.2	19.9	0.77
S19	−21.0	−170	11.1	20.3	4.12
S20	−23.9	−174	13.8	16.8	5.87
S21	−22.4	−173	3.53	15.8	1.72
S22	−24.9	−97.4	3.07	27.2	4.63
S23	−25.1	−158	13.8	17.6	0.95
S24	−27.1	−158	3.99	18.4	2.32
S25	−19.8	−190	13.8	15.6	3.61
S26	−24.4	−136	7.81	21.1	13.8
S27	−21.4	−199	11.1	15.2	1.74
S28	−19.6	−198	12.6	13.9	6.46
S29	−23.6	−180	7.34	15.4	3.27
S30	−23.5	−182	7.01	15.3	3.04
S31	−23.9	−169	6.55	15.7	2.59
S32	−24.6	−170	13.4	17.1	7.64
S33	−16.7	−189	10.5	16.4	−3.82
S34	−28.0	−183	11.0	17.4	8.15
S35	−20.0	−197	11.5	15.0	8.27
S36	−25.0	−180	9.39	17.2	9.47
S37	−17.4	−105	11.0	25.8	13.3
S38	−29.4	−164	0.36	18.7	2.13
S39	−24.2	−199	11.0	14.4	−4.79
S40	−30.0	−161	0.43	18.6	0.90
S41	−23.2	−195	13.6	18.5	8.73
S42	−22.3	−194	13.7	17.5	4.97
S43	−20.6	−192	10.2	17.6	5.88
S44	−28.9	−128	−0.61	22.0	−3.92
S45	−24.2	−207	10.2	12.9	−0.35
S46	−32.3	ND	0.94	ND	−5.35

ND—not determined.

## Data Availability

Data are contained within the article and Appendix A.

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
