# Peer review of "Determining the Authenticity of Spirulina Dietary Supplements Based on Stable Isotope and Elemental Composition"

_foods, 2023, doi:10.3390/foods12030562_

Round 1

Reviewer 1 Report

The manuscript is well documented and clearly written. I would recommend it to be revised.

Line 258: I don’t understand the expression “For this analysis 46”. Please reformulate.

Line 272: If Br, Fe, Zn and Cl are written as symbols, please write Rubidium as Rb.

Line 297: Please reformulate the sentence: “Another parameter S4, S26 and S2 have in common are the high δ15N values”.

In manuscript, please mention the number of investigated elements (13).

Author Response

See attachement.

Reviewer 2 Report

The work reported in the paper entitled "Determining the authenticity of Spirulina dietary supplements based on stable isotope and elemental composition" provides meaningful information. However, some issues have to be addressed by the authors.

Grammar, typo and syntax should be carefully checked throughout the manuscript and corrected accordingly. For example:

Line 66: “but also but also elemental” should be “but also elemental”

Line 194 (Table 2): “13C/12C, 2/1H, 34S/32S, 18/16O, 15N/14N isotope ratio (‰)” should be “13C/12C, 2H/1H, 34S/32S, 18O/16O, 15N/14N isotope ratio (‰)”

Line 196-197: “(interquartile range (IR): –26.0 to –21.8‰).” could be “(IR: –26.0 to –21.8‰).”

Line 417: “and Portugal (n = 2),.” should be “and Portugal (n = 2).”

The introduction mentions isotopes of nitrogen, carbon, elements and then continues with the isotopes of sulfur, hydrogen and oxygen. Organize all isotope information and then mention the elements (enlarge element information).

Results and discussion:

Nowhere in this section does it mention where the results of the elemental composition are found (Table S1). Could you spend a few sentences discussing the results obtained before the multivariate analysis?

In section “3.1 Isotopic composition of Spirulina food supplements from the Slovenian market” is difficult to follow the discussion of the results since Table 1 must be considered to determine the origin of the sample, Table 2 to obtain the value of the said sample and in the text sometimes the origin is mentioned or sometimes it is indicated as Snumber. One way to make it easier for the reader could be when the origin is mentioned in the text, the sample number could be placed in parentheses and vice versa, when the sample number is placed, indicate the origin in parentheses.

Line 199:  S33 also has similar values to C4 plants. Some explanation?

Line 218-219: Add a reference.

Line 219-220: Add a reference.

Line 224-227: Based on what is mentioned in these lines, which samples were exposed to organic manure?

Line 232-233: Add a reference.

Line 240-241: “The lowest δ2H was determined in the NS and Chinese samples and the highest in samples with undeclared origin.” The lowest in the NS and the highest in undeclared origin? Is the same?

Line 245-250: Confusing wording. What do you mean by comparable? Did you perform any statistical analysis between the two curves?

Figure 1: It would be interesting to graph the line of meteoric water in another color.

Figure 2: The samples could be shown in a different color, depending on their origin. Or at least those of unknown origin to visualize similarities with other samples.

Line 272:Rubidium” replace by “Rb”.

Line 302: What does Zarrouk medium contain? Is its isotopic profile characterized?

Line 314-317: Br and Rb values do not seem close to the Hawaiian samples.

Line 324-326: Check the figure caption

Could you mention something from the third group? (Figure 2a). It only mentions S18, S9 and S44

In this section, "higher" or "significantly higher" is mentioned. Was any statistical analysis performed to compare the values of an element between samples? Statistical analysis would be useful.

There are samples whose place of origin is unknown and no characteristic or similarities was mentioned (for example: sample 20,21,28,39,42 and 45)

In section “3.2.2 Discriminant Analysis of Spirulina Samples from China, Taiwan, India and Portugal”, why do you only use data from these countries? what about the others?.

Author Response

See attachement.

Reviewer 3 Report

This manuscript investigates the feasibility of using stable isotopes and minerals content in authenticating the Spirulina dietary supplements. While this type of study is always commendable as it is time-consuming and challenging, there are a few questions for the authors in order to improve the content and readability of this study.

1) General comment: There are some minor syntax errors, please check through the entire manuscript. 

2) Abstract: line 14, avoid using "we, he, she, I" in the manuscript.

Line 20-23, there is a classification/validation using blind samples? But the data are not shown in the content in results & discussion?

3) Introduction: Would be good to improve on the objectives and the rationale of study to highlight its novelty and impact.

4) Materials & methods: For sampling, any reliable method to verify if the samples are genuine? This is especially important when you were to construct a model using them as the training set. Or else the testing set cannot be classified accurately.

Any reason for the authors to identify as non-EU (based on labelling?) As non-EU could be used to cover other countries as listed in the text. 

5) Results & discussions: The discussions on the macroelements may need improvement. The discussions given are slightly general and it did not justify well how they differ according to origin. The authors may propose the reasons that lead to the differences according to the unique environment in respective country. 

Line 272, why not standardizing the use of chemical name in symbol? 

For multivariate analysis, any reason of not using PLS-DA or OPLS-DA that can give a more accurate results, and the potential biomarkers could be identified with the highest values of VIP suggested? Justify.

Can the model built be used in determining the samples of a mixed sources (i.e mixture of different spirulina species)?  

6) References: Quite a number of them are old references, would be good to refer to more recent publications. 
